# Electrochemical Aptasensor with Antifouling Properties for Label-Free Detection of Oxytetracycline

**DOI:** 10.3390/s24175488

**Published:** 2024-08-24

**Authors:** Dimitra Kourti, Georgia Geka, Lidia Nemtsov, Soha Ahmadi, Anastasios Economou, Michael Thompson

**Affiliations:** 1Laboratory of Analytical Chemistry, Department of Chemistry, National and Kapodistrian University of Athens, Panepistimiopolis Zografou, GR-15771 Athens, Greece; dimkourti96@gmail.com (D.K.); georgia.geka101@gmail.con (G.G.); aeconomo@chem.uoa.gr (A.E.); 2Department of Chemistry, University of Toronto, 80 St. George St., Toronto, ON M5S 3H6, Canada; lidia.nemtsov@mail.utoronto.ca (L.N.); soha.ahmadi@mail.utoronto.ca (S.A.)

**Keywords:** aptasensor, oxytetracycline, electrochemical biosensor, linker, α-lipoic acid

## Abstract

Oxytetracycline (OTC) is a widely employed antibiotic in veterinary treatment and in the prevention of infections, potentially leaving residues in animal-derived food products, such as milk, that are consumed by humans. Given the detrimental effects of prolonged human exposure to antibiotics, it has become imperative to develop precise and sensitive methods for monitoring the presence of OTC in food. Herein, we describe the development and results of a preliminary label-free electrochemical aptasensor with antifouling properties designed to detect OTC in milk samples. The sensor was realized by modifying a gold screen-printed electrode with α-lipoic acid–NHS and an amine-terminated aptamer. Different electrochemical techniques were used to study the steps of the fabrication process and to quantify OTC in the presence of the Fe(CN)_6_^4−^/Fe(CN)_6_^3−^ redox couple The detectable range of concentrations satisfy the maximum residue limits set by the European Union, with an limit of detection (LOD) of 14 ng/mL in phosphate buffer (BP) and 10 ng/mL in the milk matrix, and a dynamic range of up to 500 ng/mL This study is a steppingstone towards the implementation of a sensitive monitoring method for OTC in dairy products.

## 1. Introduction

In recent years, the use of antibiotics in veterinary medicine has significantly increased, resulting in higher levels of antibiotic residues in animal-derived products, such as milk, and contributing to antibiotic resistance. The consumption of milk containing antibiotic residues may pose potential health risks to consumers, including allergic reactions and reduced effectiveness in treating infections due to drug resistance [1]. Antibiotics are naturally occurring low-molecular-weight metabolites, produced by bacteria or fungi, that either kill or inhibit the growth of other microorganisms [2]. They are classified into various groups depending on their chemical structure and mechanism of action, including ß-lactams, aminoglycosides, anthracyclines, (fluoro)quinolones, tetracyclines, lincosamides, and sulfonamides [3].

Oxytetracycline (OTC) is an antibiotic of the tetracycline group commonly used for treating infections in both animals and humans. To protect consumers and ensure food safety, the European Union has established maximum residue limits (MRLs) for antibiotics in foodstuffs of animal origin; the MRL for OTC in milk is set at 100 ng/mL [4]. To mitigate the risk of exposure to OTC, several methods for its detection have been developed. Among these, biosensors have emerged as promising lower-cost and more rapid alternatives to conventional detection methodologies such as chromatography [5]. Various types of biosensors have been explored, including electrochemical [6], optical [7], and colorimetric fluorescent versions [8]. Among these, electrochemical biosensors have demonstrated significant potential due to their rapid analytical response, enhanced sensitivity, and simplicity [9]. Specifically, electrochemical aptasensors offer additional advantages, combining high flexibility, sensitivity, and selectivity, with enhanced stability compared to antibodies, along with their ability to immobilize on diverse surfaces [10]. Various platforms have been developed for antibiotic detection in milk, and especially for detecting OTC. Specifically, a limit of detection (LOD) of 5 ng/mL was achieved using thin-film gold electrodes fabricated through metal sputtering [11]. A LOD of 4.2 ng/mL was observed using nanocomposite-modified electrodes, including multi-walled carbon nanotubes (MWCNTs), gold nanoparticles (AuNPs), reduced graphene oxide (rGO), and chitosan (CS) nanocomposites to modify a glassy carbon electrode (GCE) [12]. Enhanced sensitivity with a significantly lower LOD of 0.23 ng/mL was obtained by using a GCE grafted with diazonium salt, followed by the attachment of aptamers via carbodiimide binding [13].

In complex matrices, such as milk, the detection of a target molecule can be challenging due to the presence of diverse macromolecules (proteins, lipids etc) that may interfere with the performance of the sensor. This interference can decrease the device’s sensitivity and specificity by causing fouling or non-specific adsorption at the sensor–liquid interface. It is, therefore, essential to incorporate antifouling compounds to hinder the non-specific binding of non-target molecules on the surface [14].

The main objective of this work was the development of an electrochemical aptamer-based sensor with antifouling properties for the sensitive label-free detection of OTC. This was achieved by using a gold electrode modified with α-lipoic acid–NHS [15], a potentially antifouling compound onto which an amine-terminated aptamer was immobilized via covalent bonding. The changes at different stages of the aptasensor’s preparation and detection of OTC were probed by cyclic voltammetry (CV), differential pulse voltammetry (DPV), and electrochemical impedance spectroscopy (EIS) in the presence of the Fe(CN)_6_^4−^/Fe(CN)_6_^3−^ redox couple. The fabrication of the aptasensor is described and initial experimental results are presented, suggesting that the detection of OTC at concentrations lower than the MRL set by EU can be achieved.

## 2. Materials and Methods

### 2.1. Materials

Reagents and chemicals were of analytical grade, provided by Merck (Darmstadt, Germany) and Sigma-Aldrich (Burlington, MA, USA), and used with no further purification. The amine-terminated aptamer sequence was 5′-GGA ATT CGC TAG CAC GTT GAC GCT GGT GCC CGG TTG TGG TGC GAG TGT TGT GTG GAT CCG AGC TCC ACG TG/3AmMO/-3′; the aptamer was purchased from Integrated DNA Technologies Inc. (Coralville, IA, USA). The linker, α-lipoic acid-NHS, was purchased from MedChemExpress, (Monmouth Junction, NJ, USA) and diluted in EtOH 50% (*v*/*v*). The aptamers were diluted in phosphate buffer (PB) (10 mM, pH 7.4) containing 1 mM MgCl_2_. Oxytetracycline (OTC) hydrochloride salt was purchased from Thermo Fisher Scientific (Waltham, MA, USA)); a 100 mg/L OTC stock solution was prepared in water and OTC calibrators were prepared in PB (0.01 M, pH 7.4). All electrochemical measurements were performed using a solution of 10 mM Fe(CN)_6_^4−^/Fe(CN)_6_^3−^ containing 0.5 M KCl. For the experiments in milk, 0.01 g of low-fat dried milk powder was dissolved in 10 mL PB (pH 7.4) and centrifugated for 15′ at 950 rpm. The OTC calibrators were prepared in the supernatant solution. 

### 2.2. Apparatus and Electrodes

A 1200C Series Handheld Potentiostat/Bipotentiostat equipped with the CHI440A Software (CH Instruments, Austin, TX, USA, http://www.ijcambria.com/software.htm) and an PGSTAT12 Autolab Potentiostat equipped with the GPES 4.9 Software (Metrohm, Herisau, Switzerland) were used for the electrochemical measurements. The 220AT screen-printed electrodes (SPEs) with a 4 mm diameter gold working electrode (WE), gold counter electrode (CE), and silver reference electrode (RE) were obtained from Metrohm DropSens (Herisau, Switzerland). 

### 2.3. Preparation of Aptasensors

5 μL of a 2 mM α-lipoic acid–NHS solution was drop-casted on the gold working electrode surface of the SPEs and incubated for 3 days at 4 °C in a humidity chamber. The SPEs were rinsed thoroughly with deionized water and dried under a nitrogen stream. Then, 10 μL of a 100 μM aptamer solution was drop-casted on the working electrode surface and left for 5 h at room temperature (RT) (Figure 1). Then, the electrode was washed thoroughly with deionized water to remove any excess aptamer molecules and air-dried. Finally, the SPEs were exposed to OTC solutions ranging from 25 to 500 ng/mL for 30 min. The electrochemical measurements were taken in 10 mM Fe(CN)_6_^4−^/Fe(CN)_6_^3−^ containing 0.5 M KCl. The measurement conditions for the DPV were as follows: −0.3 to +0.35 V (anodic), +0.45 to +0.3 V cathodic, scan rate 10 mV/s, step 5 mV, pulse amplitude 25 mV, modulation time 50 ms, interval time 0.5 s. The measurement conditions for the CV were as follows: −0.4 to +0.6 V, scan rate 100 mV/s. For the EIS the measurement conditions were as follows: DC potential +0.25 V, AC potential 10 mV, and frequency range 100,000 Hz to 0.1 Hz. 

## 3. Results and Discussion

To define the optimum parameters for linker binding on the surface, different concentrations (1, 2, and 4 mM) of the linker and incubation times (1 and 3 days at 4 °C) were investigated. It was found that electrodes incubated with 2 and 4 mM of the linker for 3 days exhibited the best sensitivity for OTC detection. We also evaluated the electrode modification using different concentrations of aptamer (10, 50, and 100 μΜ), and various incubation times (1 h, 2 h, 5 h, and overnight at 4 °C and at RT) with the aptamer. A 5 h incubation time at RT with 100 μM exhibited the highest sensitivity and repeatability for OTC detection.

CV, DPV and EIS were employed to study the changes in the response of the Fe(CN)_6_^4−^/Fe(CN)_6_^3−^ probe at different steps of the aptasensor modification procedure. The electrochemical measurements obtained with the selected parameters are presented in Figure 2a, Figure 2b,c and Figure 2d, respectively. The CV (Figure 2a) and DPV (Figure 2b,c) data indicate that immobilization of the linker at the bare electrode induced only a slight decrease in the peak current, followed by a further small drop after attachment of the aptamer. The EIS spectra (Figure 2d) also confirmed these findings with a gradual increase in the charge–transfer resistance after immobilization of the linker and aptamer on the electrode. These data are consistent with the assumption that attachment of the linker and the aptamer on the electrode resulted in a continuous decrease in the active electrode surface. However, the changes in the electrochemical responses were limited compared to cases where no linker was employed [11]; this is attributed to the fact that the linker is likely attached in a perpendicular, and not in a random, spatial conformation with respect to the electrode surface (as illustrated in Figure 1), causing minimal blocking of the electrode surface. The perpendicular arrangement of the linker also seemed to promote further perpendicular attachment of the aptamer, causing only a slight further drop in the mass transfer of the electrochemical probe after attachment of the aptamer. 

The potential usefulness of the linker was demonstrated by conducting DPV experiments in PB and in the milk matrix in the absence and in the presence of linker. In PB, a well-defined and clear DPV was obtained either in the absence or presence of the linker (blue traces in Figure 3a,b). However, in the milk matrix, almost complete suppression of the DPV current was observed (red trace in Figure 3a). The DPV current drop in the milk matrix was due to fouling of the electrode surface by surface-active milk components (proteins, fat etc.), which prevented effective diffusion of the electrochemical probe to the electrode surface. In contrast, in the presence of the linker, although a drop in the DPV current was observed in the milk matrix due to interference by the milk components, a measurable DPV peak (red trace in Figure 3b) was still obtained, demonstrating the antifouling properties of the linker. 

The ability of the aptasensor to detect OTC in the milk matrix was further evaluated. Figure 4a,b illustrate the anodic DPV voltammograms and EIS spectra, respectively, in the PB medium in the absence and the presence of OTC. A net decrease in the DPV peak current, and an increase in the charge-transfer resistance, were observed after incubation with OTC, indicating re-arrangement of the aptamer structure to a more random conformation induced by binding with OTC resulting in partial blocking of the electrode surface. This phenomenon actually formed the basis for the detection and quantification of OTC. In the milk matrix, and in the absence of OTC, the peak current in the DPV voltammogram was decreased and the charge-transfer resistance was increased with respect to the PB matrix (Figure 4c,d, blue traces) due to interference by the milk components, as discussed in the previous paragraph. However, a further net decrease in the DPV peak current and an increase in the charge-transfer resistance were still observed after incubation with OTC (Figure 4c,d, red traces), suggesting that OTC could still be successfully detected in milk using the aptasensor. 

The ability of the developed aptasensors to quantitatively detect OTC was evaluated by incubating the sensors with solutions containing different concentrations of OTC (0, 25, 50, 100, 250, and 500 ng/mL) in PB and in the milk matrix for 30 min at RT. The anodic DPVs were recorded and the relative current reduction, I%, was calculated as I% = (I − i_o_)/i_o_ × 100 (where i_o_ is the DPV peak current at the aptasensor in the absence of OTC and I is the DPV peak current at the aptasensor after incubation with the OTC solution). Typical anodic DPVs in PB and in the milk matrix are presented in Figure 5a,b, and the respective calibration curves (I% vs. OTC concentration) are presented in Figure 5c. The calibration curves were linearized by plotting the I% vs. the logarithm of the OTC concentration (Figure 5d). 

The sensitivity (expressed as I% over the logarithm of OTC) was −11.9% per decade of OTC concentration in PB and −14.7% per decade of OTC concentration in the milk matrix. Conditional LODs of 14 ng/mL and 10 ng/mL of OTC were calculated for the developed aptasensor in BR and in the milk matrix, respectively, using the formula LOD = 3 × SDi/S (where SDi is the standard deviation of the intercept of the respective calibration plot at low concentrations and S is the slope of the respective calibration plot at low concentrations).

## 4. Conclusions

In this work, initial results for the fabrication of a label-free aptasensor with antifouling properties for OTC detection are presented. A gold electrode was sequentially modified with α-lipoic acid–NHS and an amine-terminated aptamer. The various stages of the modification process were studied using CV, DPV and EIS with Fe(CN)_6_^4−^/Fe(CN)_6_^3−^ as a redox probe. It was demonstrated that the modification with α-lipoic acid–NHS reduced fouling of the electrode and enabled detection of OTC in the milk matrix. Proof-of-principle detection of OTC at concentrations lower than the maximum acceptable residue limit set by the EU was achieved with an LOD of 14 ng/mL in BP and 10 ng/mL in milk and a dynamic range extending up to 500 ng/mL using DPV as the detection technique. Further experiments will be performed with different types of milk and milk matrix dilutions to study the ability of the antifouling linker to prevent electrode fouling. 

## Figures and Tables

**Figure 1 sensors-24-05488-f001:**
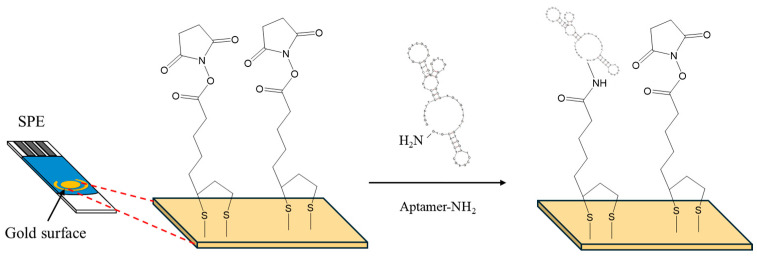
Surface modification of SPEs with α-lipoic acid–NHS, and aptamer.

**Figure 2 sensors-24-05488-f002:**
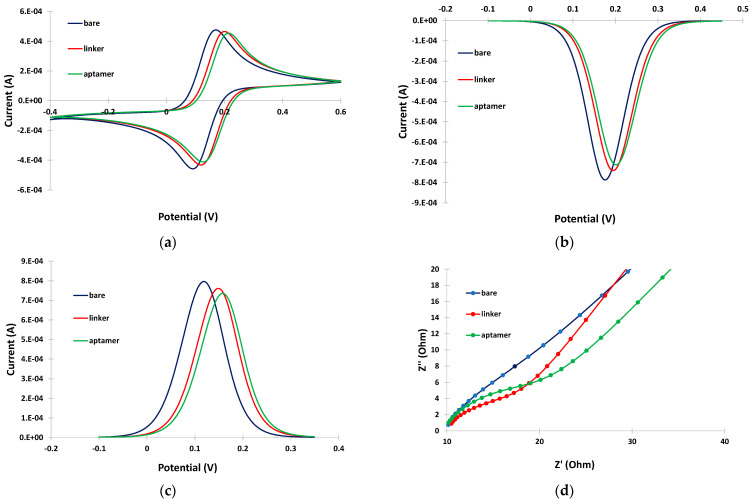
(**a**) CVs; (**b**) cathodic DPVs; (**c**) anodic DPVs; and (**d**) EIS spectra at the bare gold electrode (blue trace), after linker incubation (red trace), and after aptamer incubation (green trace). The sensors were prepared by incubation with 2 mM α-lipoic acid–NHS for 3 days at 4 °C, followed by incubation in 100 μΜ aptamer for 5 h at RT.

**Figure 3 sensors-24-05488-f003:**
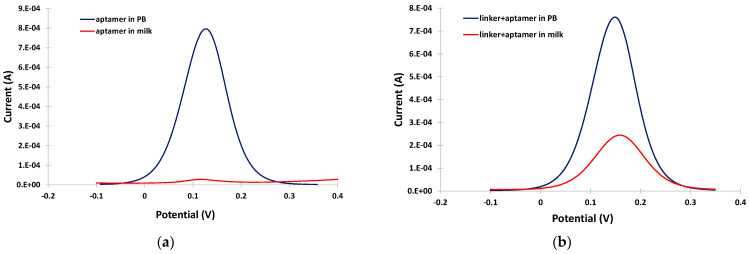
Anodic DPVs in PB (blue trace) and in milk (red trace) at the gold electrode modified (**a**) only with the aptamer; and (**b**) the linker and the aptamer.

**Figure 4 sensors-24-05488-f004:**
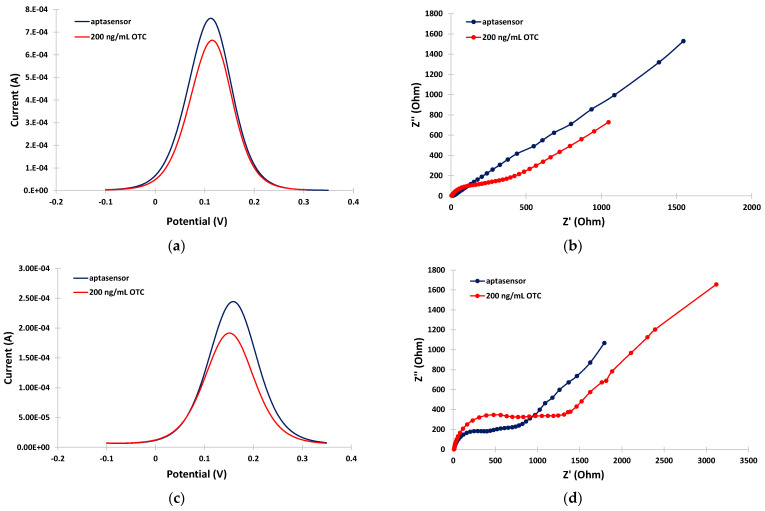
Electrochemical measurements in the absence (blue traces) and in the presence (red traces) of 200 ng/mL OTC: (**a**) anodic DPVs in PB; (**b**) EIS spectra in PB; (**c**) anodic DPVs in the milk matrix; (**d**) EIS spectra in the milk matrix. The sensors were prepared by incubation with 2 mM α-lipoic acid–NHS for 3 days at 4 °C, followed by incubation in 100 μΜ aptamer for 5 h at RT.

**Figure 5 sensors-24-05488-f005:**
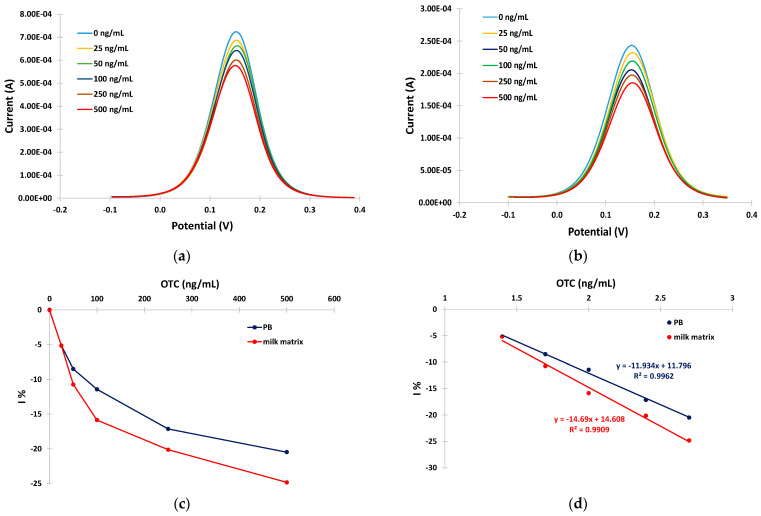
Anodic DPVs for 0, 25, 50, 100, 250, and 500 ng/mL of OTC in: (**a**) PB; (**b**) in milk matrix ((**c**) respective calibration curves in PB (blue trace) and milk matrix (red trace)); and (**d**) linearized calibration curves in PB (blue trace) and milk matrix (red trace). The aptasensors were prepared by incubation with 2 mM α-lipoic acid–NHS for 3 days at 4 °C, followed by incubation in 100 μΜ aptamer for 5 h at RT.

## Data Availability

The relevant data are available from the corresponding author upon reasonable request.

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
