# Peer review of "Electrochemical Aptasensor with Antifouling Properties for Label-Free Detection of Oxytetracycline"

_sensors, 2024, doi:10.3390/s24175488_

Round 1
Reviewer 1 Report
Comments and Suggestions for Authors
There are several concerns and questions about this short communication to be addressed:
1. What was the reason behind usage of MgCl2 for the preparation of the buffer solution? The authors are kindly asked to explain.
2. Can the authors provide the complementary EIS study of the gradual fabrication of the sensor to elucidate in more details the ongoing binding of linker, aptamer, target analyte, and perhaps some negative control sample, or the potential interferent?
3. Can the authors provide a selectivity study to rule out potential interferences? What assures the authors that the signal is specifically from the analyte?
4. The real samples study is missing.
5. Study on the non-specific binding is very important, at least some detail should be presented.
Author Response
- What was the reason behind usage of MgCl2 for the preparation of the buffer solution? The authors are kindly asked to explain.
Thanks for your question. During the preparation of the buffer solution MgCl2 was added because there is strong evidence that the presence of alkali and divalent alkaline earth cations seems to modulate the interaction between oligonucleotides and small molecules (https://doi.org/10.1021/acs.jpcb.4c02417; 10.1021/acsami.0c17535; 10.1039/D3CC04334G; https://doi.org/10.3390%2Ftoxins10110427). More specifically, as mentioned in https://doi.org/10.3390/bios14020101 Mg2+, K+ and Na+ influence the folding of DNA or RNA aptamer molecules into their three-dimensional structure, enabling them to better bind to their targets and consequently affecting their stability and binding affinity [20-22]. (22-22 are references within https://doi.org/10.3390/bios14020101)
- Can the authors provide the complementary EIS study of the gradual fabrication of the sensor to elucidate in more details the ongoing binding of linker, aptamer, target analyte, and perhaps some negative control sample, or the potential interferent?
Yes, as per your request we have added an additional figure with an EIS study of the gradual fabrication of the sensor. Figure 3D was further expanded on in the text and can be found in the revised manuscript at page 4 and 5, line 140 to 168). “[CV, DPV and EIS were employed to study the changes in the response of the Fe(CN)64-/Fe(CN)63- probe at the different steps of the aptasensor modification procedure. The electrochemical measurements obtained with the selected parameters are presented in Figures 2a, 2b-c and 2d respectively. The CV (Figure 2a) and DPV (Figure 2b-c) data indicate that immobilization of the linker at the bare electrode induced only a slight decrease of the peak current followed by a further small drop after attachment of the aptamer. The EIS spectra (Figure 2d) also confirmed these findings with a gradual increase in the charge -transfer resistance after immobilization of the linker and aptamer on the electrode. These data are consistent with the observation that attachment of the linker and the aptamer on the electrode resulted in a continuous decrease of the active electrode surface. However, the changes in the electrochemical responses are limited compared to cases where no linker was employed [11] and this is attributed to the fact that the linker is likely attached in a perpendicular, and not in random, spatial conformation with respect to the electrode surface (as illustrated in Figure 1) causing minimal blocking of the electrode surface. The perpendicular arrangement of the linker also seemed to promote further perpendicular attachment of the aptamer causing only a slight further drop in the mass-transfer of the electrochemical probe after attachment of the aptamer.]”
- Can the authors provide a selectivity study to rule out potential interferences? What assures the authors that the signal is specifically from the analyte?
Thank you for pointing this out. During the development of the sensor, each step was tested after incubation in the target analyte. The results from that study indicate that in the absence of both the linker and the aptamer, OTC antibiotic detection was not achievable. Moreover, when the sensors were tested merely in the solution used to dissolve the target analyte, i.e., water and PB buffer, the signal observed varied from the signal that was obtained in the presence of the antibiotic. Taken together, we suggest that the signal produced in the presence of analyte is specific.
As for the selectivity study, in our first submission this data was not included in attempts to keep the paper brief as this is a communication. It was our intention to elaborate on the selectivity of this sensor in a future full article related to this work. That said, our preliminary selectivity tests done in milk suggest that the sensor is selective to OTC. To address this in the text we added an additional statement on page 5, lines 194-200. “[In milk matrix and in the absence of OTC, the peak current in the DPV voltammogram was decreased and the charge-transfer resistance was increased with respect to the PB matrix (Figures 4c and 4d, blue traces) due to interference by milk components, as discussed in the previous paragraph. However, a further net decrease of the DPV peak current and an increase of the charge-transfer resistance were still observed after incubation with OTC (Figures 4c and 4d, red traces) suggesting that OTC could still be successfully detected in milk using the aptasensor.]”
- The real samples study is missing.
Agreed. As this paper is limited to a communication, the intention was to showcase the sensor as a proof-of-concept. Thus, real sample studies were omitted. However, after some consideration we have added a real sample study where some first-trial testing of the sensor was carried out in milk. Figures 3, 4 and 5 now contain milk sample studies. This was largely addressed in the Results and Discussion section of the paper.
- Study on the non-specific binding is very important, at least some detail should be presented.
Understandably, studying the non-specific binding is a necessary component of the complete sensor development. At the current state of the sensor, we believe the above mentioned preliminary testing carried out in milk samples is sufficient. However, this communication will be followed by an article where non-specific binding will be addressed in more detail.
Reviewer 2 Report
Comments and Suggestions for Authors
In the paper titled “Electrochemical Aptasensor for Label-Free Detection of 2 Oxytetracycline” was considered carefully. Although the subject is worthy of investigation, there are some technical problems, which make the paper does not meet the journal criteria. Thus, this paper may suitable for publication after revision. Some comments:
1. The Abstract needs to be improved.
2. Some annotations need to be revised, such as the picture of the surface modification of SPEs with α-lipoic acid-NHS, and aptamer should be labeled as Figure 1. And Figure 3. (a) missing half bracket.
3. What are the advantages of this sensor?
Author Response
- The Abstract needs to be improved.
Thank you for pointing this out. We agree with your comment and have thus updated the abstract to read the following on page 1, abstract and lines 13-23. “[Oxytetracycline (OTC) serves as a widely employed antibiotic in veterinary treatment and prevention of infections, potentially leaving residues in animal-derived food products that are consumed by humans, such as milk. Given the detrimental effects of prolonged human exposure, it becomes imperative to develop precise and sensitive methods for monitoring the presence of OTC in food. Herein, we present the development and results of a preliminary label-free electrochemical aptasensor with antifouling properties designed to detect OTC in milk samples. The sensor was realized by modifying a gold screen-printed electrode with α-lipoic acid-NHS and an amine-terminated aptamer. The detectable concentrations satisfy the residue limits set by the European Union with a LOD of 14 ng/mL and a dynamic range of up to 500 ng/mL. This study is a stepping stone to the implementation of a sensitive monitoring method for OTC in dairy products.]”
- Some annotations need to be revised, such as the picture of the surface modification of SPEs with α-lipoic acid-NHS, and aptamer should be labeled as Figure 1. And Figure 3. (a) missing half bracket.
Thank you for pointing this out! We have made the appropriate changes.
- What are the advantages of this sensor?
The advantages of this sensor can be attributed various factors. The sensor meets the guidelines for detection set by the European Union which suggests its capability for commercialization. Moreover, with the use of the linker detection in milk samples was accomplished, this is a great achievement as milk typically fouls the surface. Moreover, relative to other methods used in industry the approach of an electrochemical aptasensor enables a quick and accurate response. Finally, the fabrication protocol for the sensor is easy and a cheaper alternative.
Reviewer 3 Report
Comments and Suggestions for Authors
The study titled “Electrochemical Aptasensor for Label-Free Detection of 2 Oxytetracycline” has been submitted for publication by the authors. In order for this work to be fit for publication, the following comments should be addressed.
1. The title does not give the reader any detail of the samples that were analysed, it is clear that the authors are to design an Electrochemical Aptasensor for Label-Free Detection of 2 Oxytetracycline, but in what samples? This information is critical to the readers.
2. The abstract does not give any information of interest to readers. The abstract should describe what is your paper about, why is it important? How was it done? What did you find? why are your findings important? What conclusion and recommendation can you draw from this study? None of these were covered on the abstract. I advise the authors to rewrite the abstract and give cover all these key questions.
3. Electrochemistry cannot be used as a keyword, it is a field of study, not what was done in this study. Also, linker should be replaced with a suitable keyword.
4. More work on OTC sensors designed and applied for the detection of OTC in milk and other applications could be discussed in the Introduction.
5. Figure 2 a and b show that the Bare electrode’s response was better that that modified with the linker and aptamer. Then, this means that the linker and aptamer does not improve the sensitivity of the electrode. Then why were subsequent experiments carried out electrodes that were less sensitive?
6. Also, in Figure 3a, the zero OTC concentration solution shows a higher current response than the highest concentration solution. How can the authors explain this?
7. What was the electrochemical surface area of the sensor before and after it was modified? How did this influence the performance of the sensor? What about the conductivity of the sensor what happened to it before and after modification?
8. What was the sensitivity of the sensor?
9. Why were the interference studies not carried out to evaluate the sensors effectiveness in the presence of other analytes? This is very critical.
10. The stability of the sensor was not evaluated which brings a question on the stability of the sensor.
11. Three things are important when designing a sensor: Sensitivity, Stability and Selectivity. The authors should evaluate the 3S’s and present the results on the paper.
12. The analytical figures of merits are critical for sensors as well. In this study, only the LODs were evaluated, but what about the %RSD, interday and intraday precision to evaluate the reproducibility of the sensor, as well as LOQ if the sensor is designed to quantify the OTC.
13. What was the deposition potential, deposition time and pH optimum values used? These are critical factors in DPV analysis.
14. The authors should compare their results with those reported previously in a Table format, with references, and discuss their results against those they compare with.
15. How does this sensor perform in real samples, especially the milk samples that the authors discuss in the introduction. Can this sensor be deployed in a real-world application? The response to this question should be backed by evidence.
16. The conclusion does not take the readers into confidence. Was the aim of the study achieved? What evidence show that the aim was achieved? The authors should restate their main points and provide closure. They may also offer suggestions on how the research can be expanded or improved.

Author Response
- The title does not give the reader any detail of the samples that were analysed, it is clear that the authors are to design an Electrochemical Aptasensor for Label-Free Detection of 2 Oxytetracycline, but in what samples? This information is critical to the readers.
Thank you for your comment. We have emphasized the samples used in the abstract, we believe it is more appropriate to be mentioned there. However, we have changed the title to include the antifouling capabilities of the sensor. It now reads “Electrochemical Aptasensor with Antifouling Properties for Label-Free Detection of Oxytetracycline.”
- The abstract does not give any information of interest to readers. The abstract should describe what is your paper about, why is it important? How was it done? What did you find? why are your findings important? What conclusion and recommendation can you draw from this study? None of these were covered on the abstract. I advise the authors to rewrite the abstract and give cover all these key questions.
We agree with your comment and have thus updated the abstract to read the following on page 1, abstract and lines 13-22. “[Oxytetracycline (OTC) serves as a widely employed antibiotic in veterinary treatment and prevention of infections, potentially leaving residues in animal-derived food products that are consumed by humans, such as milk. Given the detrimental effects of prolonged human exposure, it becomes imperative to develop precise and sensitive methods for monitoring the presence of OTC in food. Herein, we present the development and results of a preliminary label-free electrochemical aptasensor designed to detect OTC in milk samples. The sensor was realized by modifying a gold screen-printed electrode with α-lipoic acid-NHS and an amine-terminated aptamer. The detectable concentrations satisfy the residue limits set by the European Union with a LOD of 14 ng/mL and a dynamic range of up to 500 ng/mL. This study is a stepping stone to the implementation of a sensitive monitoring method for OTC in dairy products.]”
- Electrochemistry cannot be used as a keyword, it is a field of study, not what was done in this study. Also, linker should be replaced with a suitable keyword.
Thank you for your comment. We have replaced electrochemistry with electrochemical biosensor. As for linker, we suggest keeping this word in the abstract because facilitates other biosensor researchers to find articles that used a linker to immobilize an aptamer on the surface. Other words such as compatibilizer or coupling agent are inaccurate in describing the α-lipoic acid-NHS used in this work. – Page 1, key words, line 23.
- More work on OTC sensors designed and applied for the detection of OTC in milk and other applications could be discussed in the Introduction.
As this is a communication, we felt that the following lines sufficiently discuss the detection of OTC, lines 36-71. Thank you.
- Figure 2 a and b show that the Bare electrode’s response was better that that modified with the linker and aptamer. Then, this means that the linker and aptamer does not improve the sensitivity of the electrode. Then why were subsequent experiments carried out electrodes that were less sensitive?
Thank you for your comment. Figure 2 a and b demonstrate the electrochemical response at each step of the modification on the initially bare gold electrode. This does not mean the sensitivity decreased after modification. Alternatively, this suggests that the desired chemical was immobilized on the surface, respectively α-lipoic acid-NHS and aptamer. Notably, this figure does not include any testing of OTC, hence sensitivity towards detection of OTC is not addressed in this figure. OTC detection is addressed in Figure 3. Figure 3 shows that as the concentration of OTC increases the current signal decreases, this is a typical finding for biosensors. Please note, since we have added more figures the figure numbers changed, Figure 3 is now part of Figure 5.
- Also, in Figure 3a, the zero OTC concentration solution shows a higher current response than the highest concentration solution. How can the authors explain this?
Agreed! This is an expected result. As the concentration of OTC increases, there is more interaction between the target analyte and the aptamer. At the highest concentration, the electrode surface becomes maximally covered, resulting in the greatest reduction in the electrochemical signal. The reduction can be attributed to the decreased exposure of the electrode surface to the redox probe, the Fe(CN)64-/Fe(CN)63- electrolyte solution.
- What was the electrochemical surface area of the sensor before and after it was modified? How did this influence the performance of the sensor? What about the conductivity of the sensor what happened to it before and after modification?
We appreciate your questions. As mentioned in the paper, the working electrode has an area of 4 mm in diameter. We suspect that immobilization of the aptamer may alter the surface area of the electrode. However, as this paper is limited to a communication with focus on preliminary findings, a detailed discussion on how surface area changes impact sensor performance will be provided in a subsequent paper with more comprehensive data.
- What was the sensitivity of the sensor?
Thanks for asking. The LOD or analytical sensitivity of the sensor is 14 ng/mL. This is mentioned on line 170, page 5.
- Why were the interference studies not carried out to evaluate the sensors effectiveness in the presence of other analytes? This is very critical.
We agree with you! In the interest of keeping this paper as a communication, we felt interference studies should be saved for a subsequent paper that included antifouling studies, potentially with the use of blocking agents or a linker that prevents non-specific adsorption. For this paper, a proof-of-concept sensor was desirable. Having said that, and mentioned above, we have since added some preliminary real sample results to further demonstrate the capabilities of this sensor in milk samples. This can be found in Figures 3, 4 and 5.
- The stability of the sensor was not evaluated which brings a question on the stability of the sensor.
Thank you for pointing this out. The stability of the sensor is undeniably important, especially when considering commercialization. However, at this stage, the sensor presented in the paper is merely a proof-of-concept. Based on our experience with the linker, we suspect that the surface remains stable for at least a few days. Further testing will be conducted to evaluate the long-term stability of the sensor with the aptamer immobilized on the surface, which will be addressed in future works from this collaboration.
- Three things are important when designing a sensor: Sensitivity, Stability and Selectivity. The authors should evaluate the 3S’s and present the results on the paper.
The 3S’s are important when designing a sensor. In this paper, we have addressed sensitivity, and selectivity with real milk samples. Our experience with these sensors suggests an adequate level of stability however, a comprehensive study of the sensors’ stability will be carried out and included in a subsequent paper that focuses on further improving the sensor. Thank you for your comment.
- The analytical figures of merits are critical for sensors as well. In this study, only the LODs were evaluated, but what about the %RSD, interday and intraday precision to evaluate the reproducibility of the sensor, as well as LOQ if the sensor is designed to quantify the OTC.
Absolutely, as mentioned this paper is a communication and thus, we showcase only preliminary of our collaboration. These factors will be addressed in future studies of this sensor. Thank you.
- What was the deposition potential, deposition time and pH optimum values used? These are critical factors in DPV analysis.
Indeed, DPV does not involve deposition time and potential. Only the stripping variants of DPV (i.e stripping voltammetry in which accumulation of analytes on the working electrode is used) make use of deposition time and potential. The solution used for DPV experiments in ferri/ferro in 0.5 M KCl, a medium almost universally used in electrochemical aptasensors. No particular study of pH is necessary for this solution.
- The authors should compare their results with those reported previously in a Table format, with references, and discuss their results against those they compare with.
Thank you for the suggestion. We intend to complete a table format comparison in a larger paper which includes the use of fabricated electrodes. As this is a communication, we are restricted to what has been presented in this paper along with the addition of real sample studies.
- How does this sensor perform in real samples, especially the milk samples that the authors discuss in the introduction. Can this sensor be deployed in a real-world application? The response to this question should be backed by evidence.
Thanks for your comment, our newly added milk sample studies reveal that the sensor can detect OTC, this has been added to the results and discussion section of the paper. With the use of the linker, the milk did not foul the surface of the sensor and OTC was detectable. Moreover, as described in the paper the sensor is designed with EU guidelines in mind as the goal is to detect OTC in industrial milk samples.
- The conclusion does not take the readers into confidence. Was the aim of the study achieved? What evidence show that the aim was achieved? The authors should restate their main points and provide closure. They may also offer suggestions on how the research can be expanded or improved.
Our conclusion summarizes the paper by stating how the sensor was made, the LOD, dynamic range, antifouling properties and expands on the findings that the sensor meets EU guidelines. Moreover, over in our last sentence we explain which experiments will be carried out in the future. Thank you.
Round 2
Reviewer 1 Report
Comments and Suggestions for Authors
The changes made by authors have increased the quality and clarity of the article. Furthermore, the responses to my questions from the first revision (considering the buffer preparation, gradual fabrication of the sensor, selectivity issues, non-specific binding, and real samples) as well as the added results and discussion were comprehensive and addressed all my concerns adequately. I suggest that the article is accepted for publication in current state.
Author Response
Dear Reviewer, thank you for your suggestions to improve the paper!
Reviewer 3 Report
Comments and Suggestions for Authors
1, The authors should calculate the sensitivity of their sensors.
2. Provide a comparison Table comparing their work to any related previous work and discuss the impact of their work based on this Table.
3. The conclusion should be revised, more details should be included on the conclusion, particularly on the electrochemistry of the developed sensor. Was the aim and objectives achieved? This should be highlighted fully in the conclusion.
Author Response
- The authors should calculate the sensitivity of their sensors.
The sensitivity is expressed as the ratio of the response over the concentration of the target analyte. In this case, the sensitivity is not uniform across the calibration range since the calibration plot is not linear. This is why the response was linearized by plotting the response over the logarithm of the concentration of the target analyte. Therefore the sensitivity is expressed as I% over the logarithm of the concentration of the target analyte and its value is -11.9% per decade of OTC concentration in PB and -14.7% per decade of OTC concentration in milk matrix. This is added in the manuscript.
- Provide a comparison Table comparing their work to any related previous work and discuss the impact of their work based on this Table.
This is a short communication in which no systematic optimization of the methodology was undertaken. Therefore, we think that a comparison is not appropriate in this case.
- The conclusion should be revised, more details should be included on the conclusion, particularly on the electrochemistry of the developed sensor. Was the aim and objectives achieved? This should be highlighted fully in the conclusion.
The conclusions section was revised.